# Application of the J-Integral and Digital Image Correlation (DIC) to Determination of Multiple Crack Propagation Law of UHPC under Flexural Cyclic Loading

**DOI:** 10.3390/ma16010296

**Published:** 2022-12-28

**Authors:** Yanfei Niu, Junqi Fan, Xiaoyan Shi, Jiangxiong Wei, Chujie Jiao, Jie Hu

**Affiliations:** 1School of Civil Engineering, Guangzhou University, Guangzhou 510640, China; 2Institute of Defense Engineering, AMS, Chinese People’s Liberation Army, Luoyang 471023, China; 3School of Materials Science and Engineering, South China University of Technology, Guangzhou 510640, China

**Keywords:** UHPC, fatigue crack propagation behavior, Paris law, J-integral

## Abstract

This study investigated the fatigue crack propagation behavior of ultra-high-performance concrete (UHPC) incorporated with different steel fiber lengths of 6, 13, and 20 mm under flexural cyclic loading, based on the Paris law and nonlinear fracture mechanics. In addition, multiple crack covering areas and fatigue J-integral amplitudes were employed to quantitatively evaluate the fatigue crack propagation rate and predicate the fatigue life of the UHPC during the steady development stage. The results indicated that the maximum crack opening displacement (COD) values were 0.312, 0.673, and 1.265 mm and the minimum crack growth rates were −3.05, −4.48 and −4.62 for SF6, SF13, and SF20, respectively. The critical crack length was approximately 65 mm for UHPC specimens containing different fiber length at a given fiber volume fraction (2.0%), indicating that the critical crack length was simply related to the fiber length. Interestingly, when the fatigue crack area of all the tested series reached approximately 35 ${\mathrm{mm}}^{2}$, fracture failure occurred. There were very small predictions between the actual tested and predicated fatigue lives, all less than 7.21%. Hence, it was reasonable to predict the fatigue life of the UHPC based on the J-integral according to the DIC technique.

## 1. Introduction

Alongside advances in building materials, the application of ultra-high-performance concrete (UHPC) to developments in architectural structures is becoming increasingly more extensive. UHPC is a new type of ultra-high strength cement-based material with high toughness and excellent durability [1,2,3]. With these excellent properties, UHPC applications for pavement, airfield runways, road surfaces, bridge decks, and offshore structures have substantially increased over the last decade [4]. Most of these structures work under cyclic loading conditions [5,6,7,8,9,10,11,12,13,14]. Under the cyclic loading, fatigue damage will occur in architectural structures during its service process, and tiny microscopic cracks will gradually expand into macroscopic cracks, resulting in the decline of mechanical properties and durability, and seriously jeopardizing the safety and reliability of the structure [15,16]. Therefore, it is very important to investigate the crack propagation behavior and damage properties of the UHPC under cyclic loading.

The nature of fatigue damage involves crack initiation, and cumulative, and propagation processes. Zhang [17] studied the crack propagation behavior of fiber reinforced concrete (FRC) under flexural cyclic loading and found that all of the tested samples showed a similar trend, where the fatigue crack increased gradually with slight changes in the first stage of the cycles, while maintaining constant increase rate in the second stage and then ascending dramatically up to failure. V.C. Li [18] observed that the rapid development stage involved matrix crack initiation, a steady stage relating to the formation of microcracks, and an accelerated stage correlating to the combination of microcracks and macrocracks. Horii [19] divided fatigue crack growth into two distinct stages: the deceleration stage, where the fatigue crack growth rate decreased as the cracks increased, and the acceleration stage, where the fatigue crack growth rate solidly grew until failure. Generally, most researchers believe that the fibers can obviously improve the flexural fatigue performance of the FRC by inhibiting crack growth [20]. However, Hsu [21] suggested that the incorporation of fibers can more benefit the promotion of fatigue performance under low-cyclic loading than high-cyclic loading. Jalkh [22] also agreed with Hsu, and insisted that fiber bridging and pullout dissipated a significant amount of energy, releasing the strain concentration in the crack tip under lower stress levels than higher stress levels. Various studies have been conducted on the fatigue of the FRC to describe the fatigue crack propagation behavior and failure mechanism. Nevertheless, there does not seem to be a comprehensive quantitative appreciation of fatigue crack propagation behavior based on fracture mechanics.

The application of fracture mechanics provides a promising method for elucidating the fatigue crack propagation mechanism involved in nonlinear fracture mechanics (NLFM) and Paris law [23], which describes the relationship between the fatigue crack propagation rate (da/dN) and the crack tip stress intensity factor amplitude (ΔK). Bazant [24] investigated the size of the effect on concrete beams based on the Paris law combined with NLFM. Mohamadreza [25] developed a calibrated finite element model using concrete damage plasticity to predicate the behavior of UHPC. Charmistha [26] employed the Paris law to assess the fatigue performance of concrete structures based on a nonlinear crack propagation model. Tanaka [27] studied the effect of fiber orientation on the fatigue crack propagation behavior, where the edge notched beams were adopted by Paris law, and the study concluded that ΔK played a major role in controlling the fatigue crack propagation rate at low cycles and the maximum crack tip stress intensity factor ($K_{\max}$) at high cycles. Carlesso [28] found that the envelope curve of high-performance fiber-reinforced concrete (HPFRC) under flexural cyclic loading could be approximated to the static flexural loading- crack opening displacement (COD) curves, and reported that the failure criterion could also be used to evaluate the fatigue deformation of HPFRC. Thus, employing the Paris law along with NLFM has shown to be a proven and effective method for illuminating the fatigue crack propagation behavior. Unfortunately, the use of Paris law to describe the evolution of fatigue crack has not been sufficiently studied. Specifically, as (I) the single edge notched will generate a stress/strain concentration zone near the crack tip, accelerating fatigue crack growth, and bringing about the differentiation between the actual and predication fatigue life of the concrete. In addition, (II) Paris law has only been applied to ordinary concrete or FRC with a single fatigue crack, making it difficult to evaluate the fatigue crack propagation behavior of tested specimens with multiple cracks. Xu [29] proposed amendments to the Paris law and introduced two parameters, namely, the crack covering area and fatigue J-integral amplitude, to estimate the fatigue multiple crack propagation behavior.

The objective of this study was to employ an amendment to the Paris law to quantitatively evaluate the fatigue multiple crack propagation behavior and predict the fatigue life of UHPC incorporated with various fiber lengths (6, 13, and 20 mm). Specifically, the temporal evolution of the crack patterns, crack opening displacement (COD), and crack length were quantitative characterized by utilizing the DIC technique. Then, the changes in the coverage area of fatigue multiple cracks with the number of cycles was obtained. Finally, the fatigue life in the steady crack development stages was predicated based on the amended Paris law by introducing the J-integral amplitude.

## 2. Experimental Program

### 2.1. Materials and Mixtures Composition

The matrix composition of the UHPC is clearly differed from ordinary concrete. The cementitious materials employed in this study were PII 52.5 Portland cement and silica fume. The detailed chemical composition of the cementitious materials is presented in Table 1. The particle size distribution of the cementitious materials is exhibited in Figure 1, which the silica fume contains 98% SiO_2_ and has an average diameter 0.2 μm. By using particle packing theory, the fine aggregate with two different sizes (0.160–0.315 mm and 0.63–1.25 mm) is adopted to increase stacking density. Superplasticizer with a solid content 30% was introduced to improve the workability for the low water to binder ratio fresh matrix. To estimate the effect of steel fiber length on the flexural fatigue properties and crack propagation behavior, straight steel fibers with identical diameter of 0.2 mm and with different fiber lengths ($L_{f}$ = 6 mm, 13 mm and 20 mm) were incorporated at a fiber volume fraction of 2.0%. The physical properties of the steel fiber are listed in Table 2 and the distributions of fiber incorporated in UHPC are shown in Figure 2. Park et al. [30,31] investigated the contribution of fiber contents to first cracking strength and flexural tensile strength, and proposed a method to optimize the mix proportion of the UHPC. While the mix proportion of the UHPC adopted in this study is shown in Table 3. The water-binder ratio of the UHPC mature was 0.16 and the binder-sand ratio was 1:1. Detailed information regarding to the mixing sequence of UHPC mixtures can be found in Niu et al. [32]. After casting, the prepared specimens were covered with plastic sheets to reduce water evaporation and cured at room temperature for 24 h prior to demolding; and then, the tested specimens were maintained into the water tank for 90 d to minimize the effect of strength increase of the UHPC during the fatigue testing.

### 2.2. Fatigue Testing

Static flexural testing and fatigue flexural testing were conducted on an MTS Landmark 370.25 servo-controlled electro-hydraulic machine made in USA with a capacity of 250 kN. To determine the maximum and minimum flexural loading ($P_{\max}$ and $P_{\min}$) for the fatigue testing, the static flexural testing was executed before the fatigue testing. The geometric dimensions of the testing beam were 100 mm × 100 mm × 400 mm with a clear span of 300 mm. Monotonic flexural testing conducted on the MTS testing machine was implemented under the displacement control with a constant rate of 0.2 mm/min. The average monotonic flexural strength ($P_{f}$) of each batch was obtained from the results of the three tested specimens. The averaged static flexural strength of UHPC incorporated steel fiber with lengths of 6, 13 and 20 mm were 15.77 (cov% = 5.5%), 20.71 (cov% = 4.2%) and 25.26 MPa (cov% = 3.4%), respectively. Fatigue testing was conducted under a sinusoidal loading with a constant amplitude at a frequency of 8 Hz. First, the loading increased linearly until the average values of the maximum and minimum flexural loading, ($P_{\max}+P_{\min}/2$); and then cycled in a sine manner within the limits of (P = $P_{\max}-P_{\min}$) until fracture failure. The stress levels (S = $P_{\max}/P_{f}$ = 0.70) and the value of the cyclic stress ratio (R = $P_{\max}/P_{\min}$ = 0.1) remained unchanged throughout the whole fatigue testing. The sample capacity for each batch was selected as eight, to improve the confidence level and reduce error. Fatigue testing was terminated when the tested specimens underwent fracture failure or reached to the upper limit of 2 × $10^{6}$ cycles.

### 2.3. Fatigue Crack Propagation Testing

Digital image correlation (DIC) consists of a stabilized, non-contact and nondestructive technique to measure the surface deformation of the target zone or region of interest (ROI), as shown in Figure 3. Two Charge coupled Device (CCD) cameras (1024 × 1024 ${\mathrm{pixels}}^{2}$) were mounted on the rigid tripod to capture the images of the speckle pattern in the ROI during loading. Two blue lights were used to eliminate the fluctuations of the environmental optical source. While the lateral surface of the tested specimen was painted with white and sprayed with a black dotted pattern within an area of 100 × 200 mm. The vertical distance between the central point of ROI and the laser dot was 500 mm, while the vertical orientation was 40°. To gather more deformation information for the ROI during each cycle loading, the acquisition frequency of the CCD camera was maintained at 10 frame/s, which was slightly higher than the frequency of sinusoidal loading frequency.

## 3. Results and Discussion

### 3.1. Evaluation of Localized Deformation

The DIC technique was used to capture the speckle changes on the tested specimen surfaces and then a numerical correlation was conducted to characterize the displacement or strain fields at the different crack propagation stages. Xu [29] divided the evolution of the maximum crack opening displacement (COD) into three stages: (I) the rapid development stage, where fatigue deformation increased dramatically and several cracks emerged during a few cycles; (II) the stable development stage, where new cracks were generated and the fracture energy dissipated, causing the geometry of the primary crack to show no obvious changes; and (III) the failure stage, where the primary cracks were localized and increased rapidly under failure, as shown in Figure 4. At the end of each stage, a representative image was selected to evaluate the evolution of the strain fields, as shown in Figure 5. Globally, the strain concentration region appeared at the zone of the maximum bending moment, which revealed the occurrence of crack clustering. With increasing steel fiber length, the area of strain concentration and the number of fatigue cracks increased. This indicated that abundant fracture energy was absorbed into initial new cracks and fatigue deformation grew slowly. Of further note, there was pronounced development of fatigue cracking toward the longitudinal axis of the tested specimens. However, with a longer fiber length, there was a more obvious tendency to form clusters of fatigue cracks during the crack propagation process. Thus, the UHPC that incorporated a length of 20 mm steel fiber exhibited excellent fatigue resistance performance under the uniform distribution precondition.

### 3.2. Assessed Fatigue Crack Propagation Behavior

#### 3.2.1. Maximum COD

The relationship between the maximum COD and cycle ratio for the UHPC with different fiber lengths at a stress level of 0.70 is presented in Figure 6. Clearly, the evolution of maximum COD showed three stages (I, II, and III) from crack initiation to fracture failure in the tested specimen under cyclic loading. Throughout the fatigue crack propagation process, the stable development stage (II) occupied a significant proportion and played a dominant role during the entire fatigue life; however, the rapid development and failure stages (I and III) accounted for a small percentage. The proportions of stages I, II and III were approximately established as 10%, 75% and 15%, respectively.

Combining the strain diagram (Figure 5) for analyzing the maximum COD extension process (Figure 6), there were a few minor strain concentration occurrences on the surface of the tested specimen, where the maximum strain on the bottom was approximately 327 µm, exceeding the strain value (236 µm) of the generating crack, which meant that the primary crack penetrated into the specimens and the other cracks did not reach a certain level where they could be identified as a crack [33]. This indicated that the rapid development of maximum COD mainly came from the primary crack extension in stage I. At the end of stage I, the average crack widths for SF6, SF13, and SF20 were 0.047, 0.058, and 0.142 mm. At stage II, the primary crack in the lower portion of the tested specimen showed a slight extension in the x-direction, but no obvious changes in the y-direction. Meanwhile, the strain diagram presented in Figure 5 revealed that a new crack initiated and developed in the middle of the tested specimen. Compared to the crack widths at the end of stage I, the maximum COD values showed increases of 0.048, 0.103, and 0.116 mm for SF6, SF13, and SF20, respectively. In stage III, the primary crack extended dramatically upward, and the maximum COD also developed exponentially. With fatigue failure, the maximum COD values were 0.312, 0.673, and 1.265 mm for SF6, SF13, and SF20, respectively. A comparison demonstrated that the longer steel fibers that bridged the crack surface could effectively improve the flexural fatigue performance of the UHPC, including the fatigue strength and maximum COD, by increasing the pullout loading and corresponding slip, due to the increase in bonding strength and area between the fiber and the matrix [34].

#### 3.2.2. Length of Fatigue Crack

The relationship between the crack length in the y-direction and the number of cycles is shown in Figure 7, for the UHPC incorporated with different steel fiber lengths. The results indicated that the fatigue crack in the y-direction initially propagated quickly, slowly afterward and then rapidly to failure, where the development trend presented an inverted S-shape. To quantitative analysis the fatigue crack evolution in the y-direction, a smooth continuous function was adopted to describe the relationship between the fatigue crack length and the number of loading cycles [35]:(1)$$a=\tau \ln\left[\frac{b-N}{{\mathrm{Nm}}_{1}-c}\right]$$

a: fatigue crack length; N: number of cycles; τ, b, $m_{1}$ and c: experimental calibration constants, which are shown in Table 4.

The nonlinear regression accuracy of all these corresponding relations exceeded 0.90, which indicated that the established mathematical function, Equation (1), could demonstrate the crack length evolution process of the UHPC under flexural loading. As demonstrated in Figure 7, the evolution of fatigue crack length for the UHPC with steel fiber lengths of 6 mm was obviously different from the other tested specimens (SF13 and SF20), due to the dramatically lower steady-state plateau, which indicated a longer duration of crack length in the y-direction with no evident changes.

#### 3.2.3. Crack Propagation Rate

Considering the propagation of fatigue crack length point of view, the crack propagation rate [18,36] could be calculated by using the first derivation of Equation (1):(2)$$\frac{\mathrm{da}}{\mathrm{dN}}=\tau \frac{c-{\mathrm{bm}}_{1}}{\left(b-N\right)\left({\mathrm{Nm}}_{1}-c\right)}$$

Figure 8 shows the relationship between the crack propagation rate (logarithmic scale) and the crack length for the UHPC at a stress level of 0.70. As shown in Figure 8, the crack propagation rate first decreased until the fatigue crack length reached the effective crack length ($a_{c}$) and then increased until fracture failure. The plots of the crack propagation rate with fatigue crack length appeared at two different changing stages: the deceleration rate and acceleration rate [23]. The SF6 specimens exhibited higher crack propagation rates than the SF13 and SF20 at a given stress level. At the stress level of 0.70, the minimum crack growth rates of the SF6, SF13 and SF20, respectively, were −3.05, −4.48 and −4.62. It was indicated that the longer steel fiber can restrict the development of the fatigue crack by increasing the effective bonding area between the fiber and the matrix at crack surfaces. The effective crack length ($a_{c}$) could be obtained when the slopes of the Log(da/dN)-a curves approached zero. The effective crack lengths ($a_{c}$) for the UHPC with fiber lengths of 6, 13, and 20 mm were 63.04, 64.54, and 68.81 mm, respectively. The critical crack length was approximately 65 mm for the UHPC specimens containing different fiber lengths at a given fiber volume fraction (2.0%), indicating that the critical crack length was simply related to the fiber length.

#### 3.2.4. Crack Propagation Law

Generally, the UHPC demonstrated excellent ductility and multiple-cracking characterization under flexural cyclic loading, as shown in Figure 5. Hence, the crack propagation law could be applied to ordinary concrete and fiber reinforced concrete, which consisted of fracture failure with a single fatigue crack, and was rather difficult to suitably apply to ductile fracture. Xu [13] proposed a new fatigue crack propagation law for ductile composites based on nonlinear fracture mechanics, which adopted a crack covering area and J-integral amplitude to describe the fatigue crack propagation rate [29], according to
(3)$$\frac{\mathrm{dA}}{\mathrm{dN}}=C{\left(\Delta J\right)}^{m}=C{\left(J_{\max}-J_{\min}\right)}^{m}$$

A: macrocracking area; N: number of cycles; ΔJ: J-integral amplitude; C and m: materials constants; $J_{\max}$ and $J_{\min}$: J-integral values relate to the maximum and minimum applied loading.

##### Area of Fatigue Cracking

To estimate the area evolution of multiple cracks in the UHPC under flexural cyclic loading, the DIC technique was used to capture the irregular crack patterns on the surfaces of the tested specimens. Khorami [31] introduced appropriate statistical procedures for developing fragility curves based on cracking data by assuming that they can be represented by Rice distribution functions. For the number of cracks, the width and length of the fatigue cracks were determined, and the fatigue area could be accurately calculated, as shown in Figure 9. The evolution of the fatigue crack area of the UHPC with the number of cyclic loads is shown in Figure 10. Three different change trends evidently emerged from these curves, which were similar to the maximum of COD. 

V.C. Li [18] reported that the composite fracture energy consisted of two parts: off-crack-plane matrix-cracking and on-crack-plane fiber bridging. As shown in Figure 10, when the steel fiber length increased from 6 to 20 mm, the evolution curves of the fatigue crack area in stage I rapidly increased and then gradually flattened, and the variations in the fatigue crack areas (ΔA) of the UHPC tested specimens of SF6, SF13, and SF20 were 6.12, 9.29, and 11.02 ${\mathrm{mm}}^{2}$. During this stage, the fracture energy was employed to generate cracks. In stage II, the fatigue crack areas (ΔA) increased gently and the slope of the curves decreased as the number of steel fiber increased, which indicated that stable crack propagation occurred. The variations in fatigue crack areas (ΔA) ewer 3.49, 4.90, and 6.66 ${\mathrm{mm}}^{2}$ for SF6, SF13, and SF20, respectively. During this process, considerable fracture energy was captured that facilitated fiber pullout from the UHPC matrix and then fiber bridging contributed primarily in restricting the propagation of fatigue cracking. For stage III, a smoother and rounder shape appeared when the steel fiber length increased and when the curves moved upward to a higher crack area. Off-crack plane matrix-cracking and on-crack-plane fiber bridging energy absorption occurred simultaneously, enlarging the cracks and pulling out the fibers. Interestingly, when the fatigue crack area of all of the tested series reached approximately 35 ${\mathrm{mm}}^{2}$, fracture failure occurred.

##### J-Integral

Based on nonlinear fracture mechanics, the relationship between the J-integral and equivalent stress intensity factor, K, could be defined as [29]:(4)$$K^{2}=\mathrm{EJ}$$

In order to analyze conveniently, the decaying response of the normalized elastic modulus of the tested samples with the number of cycles is shown in Figure 11. Clearly, the elastic modulus of the UHPC decreased sharply in the initial crack propagation and failure stages and then gently in the stable development stage. The fiber length specimen of 20 mm exhibited a lower decaying response of the elastic modulus during the fatigue test. It is noted that the elastic modulus (E) of the UHPC decreased to approximately 60% of the initialed elastic modulus ($E_{0}$) when the failure occurred, which barely affected by the fiber lengths. Therefore, it is reasonably concluded that the value of 0.6 could be as a limit value of the fatigue elastic modulus.

Caepinteri [7] employed DIC to analyze the latent critical crack path of unnotched concrete specimens under cyclic loading, and established the function of stress intensity factor (K) for unnotched concrete, according to Equation (5), which slightly differed from per-notched specimens [22]:(5)$$K=\frac{6M_{\max}\sqrt{a_{c}}}{{\mathrm{bh}}^{2}}\left[1.99-2.47\left(\frac{a_{c}}{h}\right)+12.97{\left(\frac{a_{c}}{h}\right)}^{2}-23.17{\left(\frac{a_{c}}{h}\right)}^{3}+24.80{\left(\frac{a_{c}}{h}\right)}^{4}\right]$$

For an unnotched concrete beam under four-point flexural bending, the effective crack length ($a_{c}$) [22] could be calculated by
(6)$$a_{c}=\frac{\left[\gamma^{\frac{3}{2}}+m_{1}\left(\beta \right)\gamma \right]h}{{\left[\gamma^{2}+m_{2}\left(\beta \right)\gamma^{\frac{3}{2}}+m_{3}\left(\beta \right)\gamma +m_{4}\left(\beta \right)\right]}^{\frac{3}{4}}}$$
(7)$$m_{1}\left(\beta \right)=\beta \left(0.25-0.0505\beta^{\frac{1}{2}}+0.0033\beta \right)$$
(8)$$m_{2}\left(\beta \right)=\beta^{\frac{1}{2}}\left(1.155+0.215\beta^{\frac{1}{2}}-0.0278\beta \right)$$
(9)$$m_{3}\left(\beta \right)=-1.38+1.75\beta$$
(10)$$m_{4}\left(\beta \right)=0.506-1.057\beta +0.888\beta^{2}$$

$M_{\max}$: maximum bending moment; b: thickness of beam; h: height of the beam; β = s/h, s is the span of the beam; $\gamma =B*t*E/\left(6P_{\max}\right)$, t is the thickness of the cross section; E: elastic modulus of the beam; $P_{\max}$: maximum load.

Compared to the effective crack length ($a_{c}$), as obtained by Equation (2), there was a slight difference from the values of the effective crack length ($a_{c}$), as calculated by Equation (6) for 67.08 mm for SF13 and 69.47 mm for SF20, respectively. Noticeably, there was a wider margin of deviation for the calculated values of effective crack length ($a_{c}$) for SF6 with 63.04 and 47.44 mm, which were calculated by Equations (2) and (6), and the reason for the deviation was probably because the flexural performance of the UHPC incorporated with 6 mm steel fiber ($V_{f}$ = 2.0%) showed strain softening behavior, which resulted in a decrease in loading carrying capacity after crack initiation.

As shown in Figure 12a, three different linear relationships were presented in the schematic diagram that displayed changes in the values of the J-integral relative to the fatigue crack area, and two characteristic points of crack initiation and localization, which corresponded to the two critical values $J_{\mathrm{IC}}$ and $J_{\mathrm{IF}}$. Figure 12b shows the relationship between the J-integral and applied flexural loading prior to when fracture failure occurred. Under static flexural loading, the maximum loading values for the UHPC incorporated with lengths of 6, 13, and 20 mm were 28.36, 43.81, and 61.92 kN, respectively, and the corresponding values of $J_{\mathrm{IF}}$ were 24.91, 10.27, and 1.10 kJ/m^2^. For a given cyclic stress ratio (R = $P_{\max}/P_{\min}$ = 0.1), the minimum loading ($P_{\min}$) related to its J-integral values ($J_{\min}$) were 0.01, 0.11, and 0.25 kJ/m^2^, which was much smaller than $J_{\mathrm{IC}}$. Therefore, the effect of $J_{\min}$ on the growth of the fatigue crack cloud would likely be negligible. Consequently, it was reasonable that Equation (2) could be simplified as [29]:(11)$$\frac{\mathrm{dA}}{\mathrm{dN}}=C{\left(J_{\max}\right)}^{m}$$
(12)m=n/2

C = 9.03 × 10^−6^; m = 3.12 (Pairs constant) [19].

#### 3.2.5. Crack Propagation Law Verification

For the entire fatigue crack propagation process of the UHPC, the stable development stage (II) had a proportion of 75%, which was close to the actual fatigue life. Therefore, the fatigue life prediction of the UHPC according to stage II was effective and could reasonably represent actual fatigue life, according to [13]:(13)$$\Delta N_{\mathrm{II}}=\int_{A_{1}}^{A_{2}}\frac{\mathrm{dA}}{C{\left(J_{\max}\right)}^{m}}$$

$\Delta N_{\mathrm{II}}$: fatigue life in stage II; $A_{1}$ and $A_{2}$: the crack area at the points of crack initiation and localization.

As shown in Figure 13, there were very small predictions between the actual tested and predicated fatigue lives, all less than 7.21%; hence, it was reasonable to predict the fatigue life of the UHPC based on the J-integral according to the DIC technique. The fatigue cracking area can be easily monitored in actual engineering by applicating the DIC technique, if C and m are provided, the fatigue life for UHPC of stage II could be obtained with ΔJ. Furthermore, the number of load cycles at any moment in this stage can be calculated by Equation (13).

In order to improve the reliability of the crack propagation law, a comparison between predicated and experimental results obtained from Xu [29] were considered. The experimental data of ultra-high toughness cementitious composites (UHTCC) under flexural fatigue loading are summary in Table 5 and then, substituting $J_{\max}$ and the regressive values of C and m into Equation (13), the fatigue life of the stage II were derived. Figure 14 reveals the comparison of tested and calculated life of UHTCC [29], and the line in this diagram was fitted by the calculated results. Obviously, the differences between tested results and calculated results were very small, which indicated that the crack propagation law also applied for UHTCC.

## 4. Conclusions

This study investigated the influence of steel fiber length on the fatigue crack propagation of a UHPC under flexural cyclic loading. Based on the Paris law and J-integral, a new theoretical fatigue multiple crack propagation law was employed using the DIC technique to obtain the area of multiple fatigue cracks. The following conclusions were drawn.

The critical crack length was approximately 65 mm for UHPC specimens containing different fiber length at a given fiber volume fraction (2.0%), indicating that the critical crack length was simply related to the fiber length;It is noted that the elastic modulus (E) of the UHPC decreased to approximately 60% of the initialed elastic modulus ($E_{0}$) when the failure occurred, which barely affected by the fiber length. Therefore, it is reasonably concluded that the value of 0.6 could be as a limit value of the fatigue elastic modulus;When the fatigue crack areas of all the tested specimens reached approximately 35 ${\mathrm{mm}}^{2}$, fracture failure occurred in the UHPC, which could be as one of the objective basis for the fracture failure occurrence;The fatigue life of stable stage could be computed by the crack propagation law, with the monitored cracking area by using DIC technique and the two regressed parameters based on nonlinear fracture mechanics.

## Figures and Tables

**Figure 1 materials-16-00296-f001:**
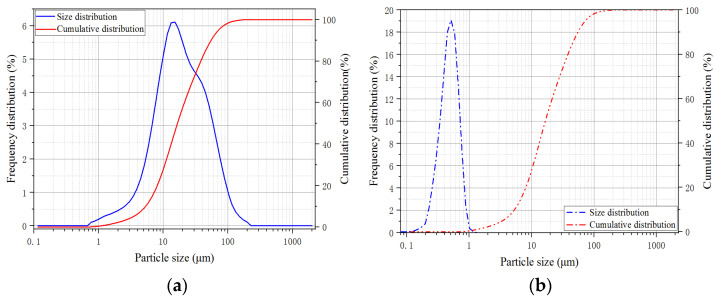
Particle size distribution of the cementitious materials. (**a**) Cement; (**b**) Silica fume.

**Figure 2 materials-16-00296-f002:**
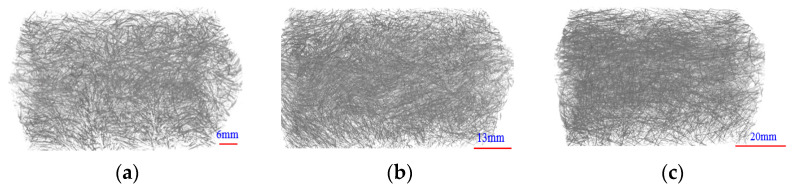
The distribution of steel fiber in UHPC. (**a**) 6 mm; (**b**) 13 mm; (**c**) 20 mm.

**Figure 3 materials-16-00296-f003:**
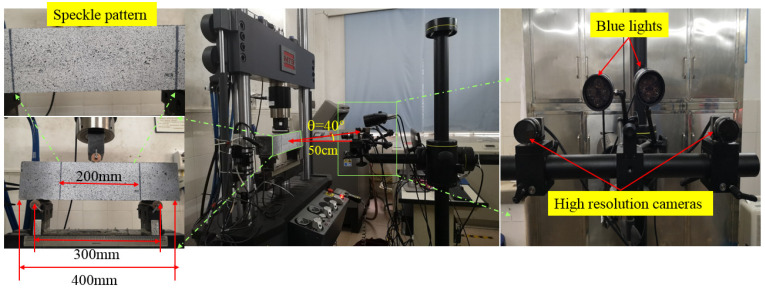
Components and application of DIC technique.

**Figure 4 materials-16-00296-f004:**
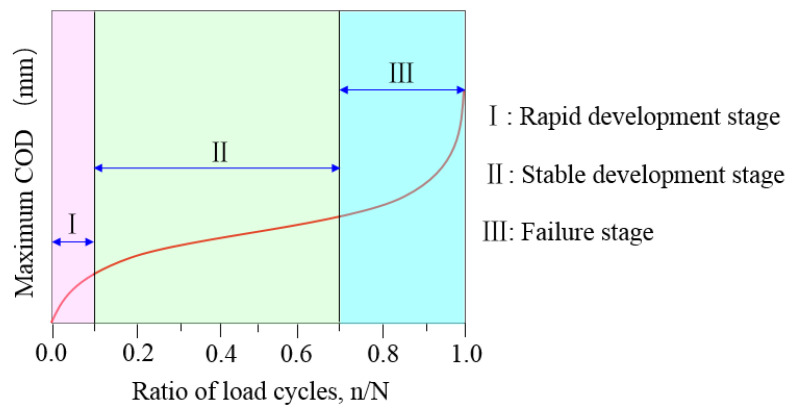
Evolution of Maximum COD with the ratio of load cycles.

**Figure 5 materials-16-00296-f005:**
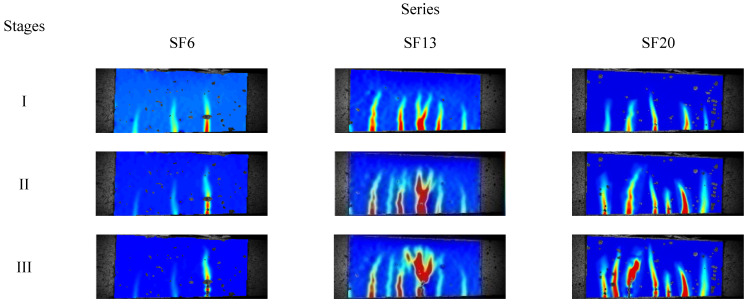
Characterization of strain fields of the region of interest by using DIC technique.

**Figure 6 materials-16-00296-f006:**
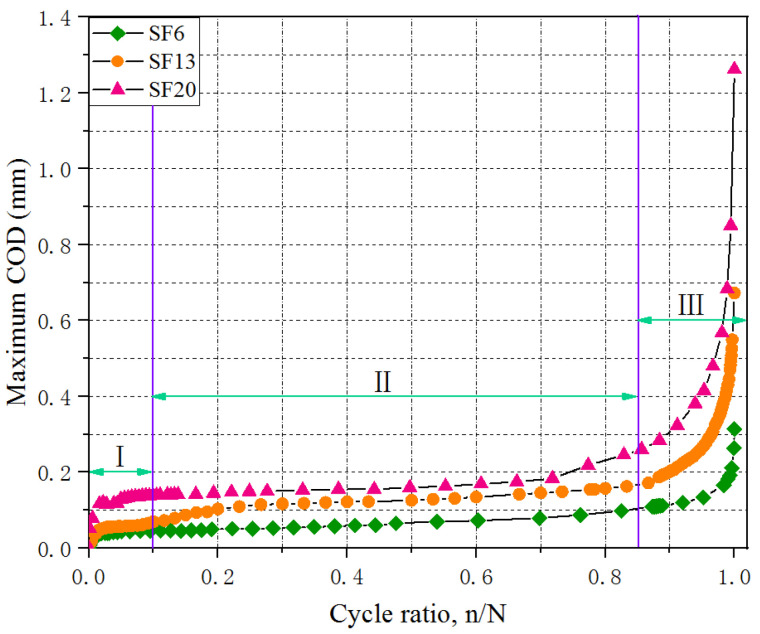
Evolution of maximum COD with the cycle ratio for UHPC with different fiber lengths.

**Figure 7 materials-16-00296-f007:**
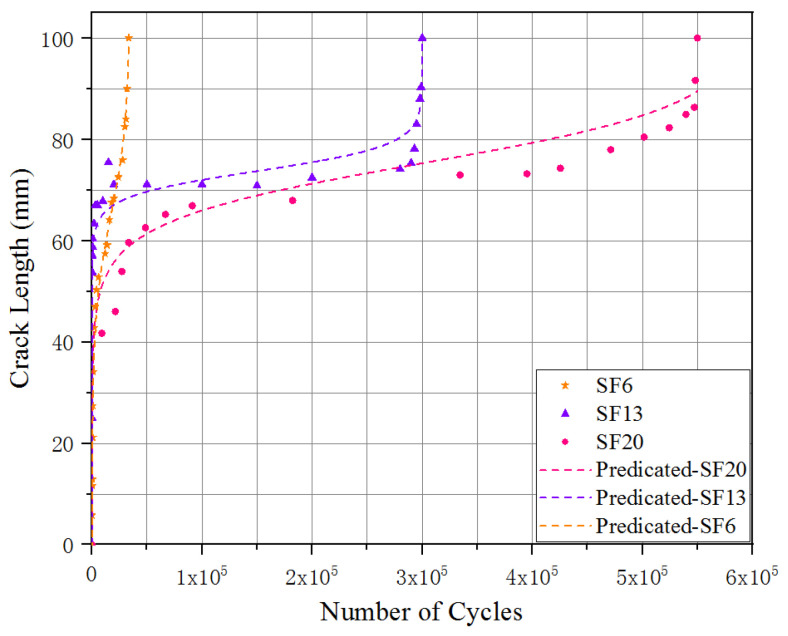
Relationship between fatigue crack length and number of cycles.

**Figure 8 materials-16-00296-f008:**
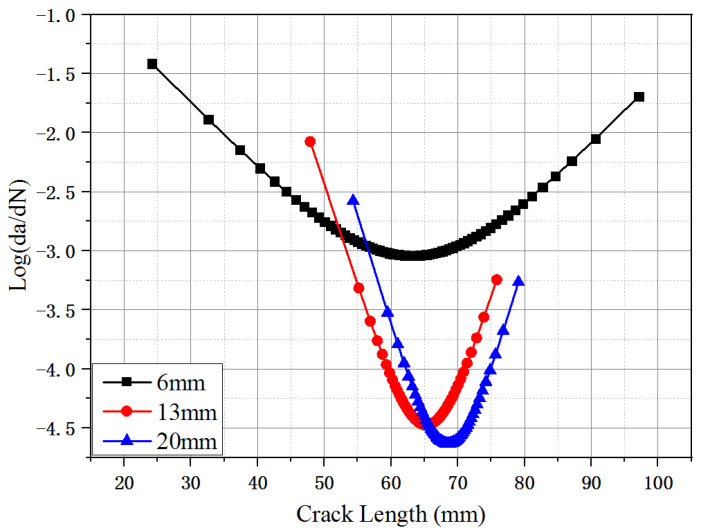
Relation between crack propagation rate (Log) and fatigue crack length.

**Figure 9 materials-16-00296-f009:**
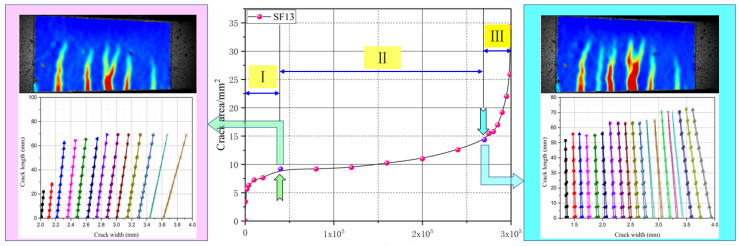
Calculating the crack area by using the DIC technique.

**Figure 10 materials-16-00296-f010:**
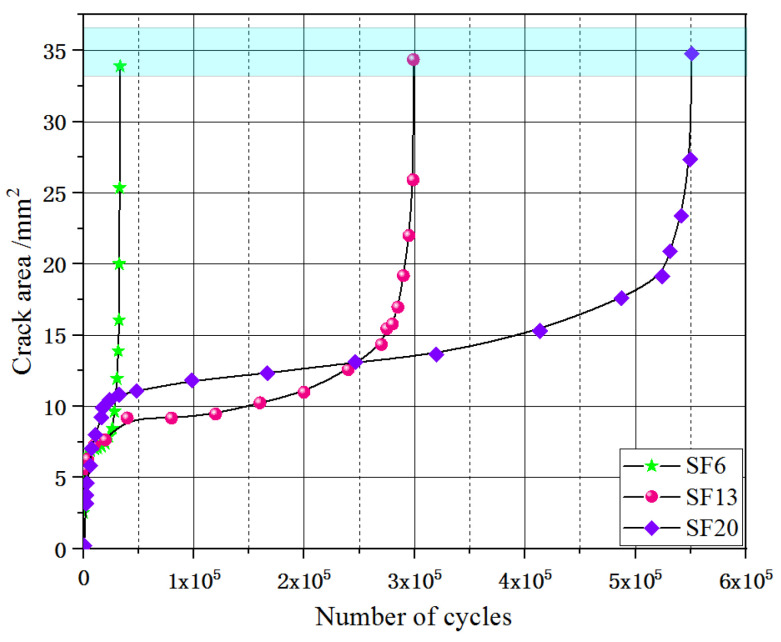
Relationship between the crack area (A) and number of cyclic loads (N).

**Figure 11 materials-16-00296-f011:**
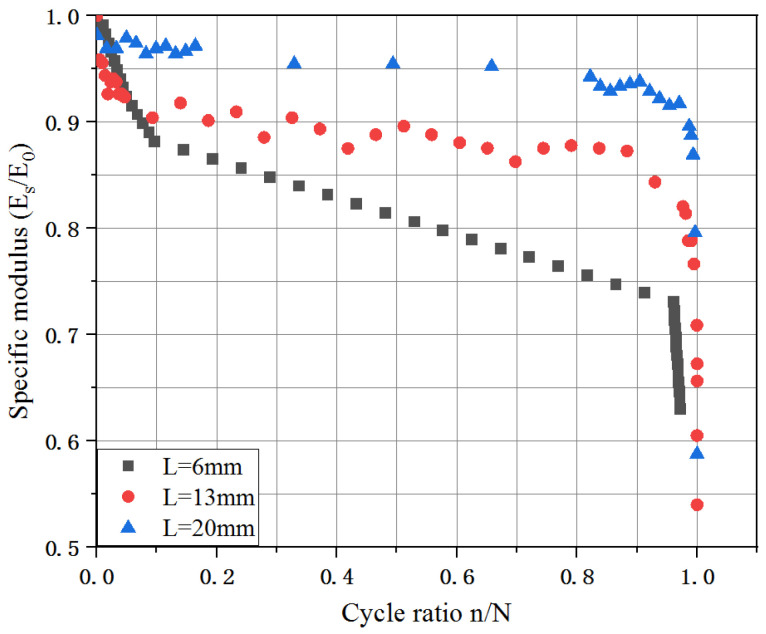
The decaying response of elastic modulus with the number of cycles.

**Figure 12 materials-16-00296-f012:**
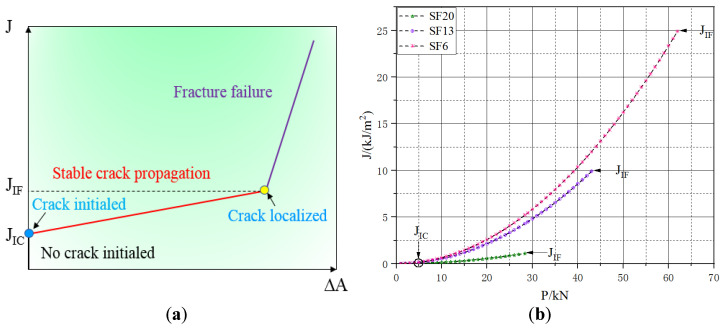
Evolution of the J-integral with (**a**) variation in crack area (Schematic diagram) and (**b**) applied load (J-integral curve).

**Figure 13 materials-16-00296-f013:**
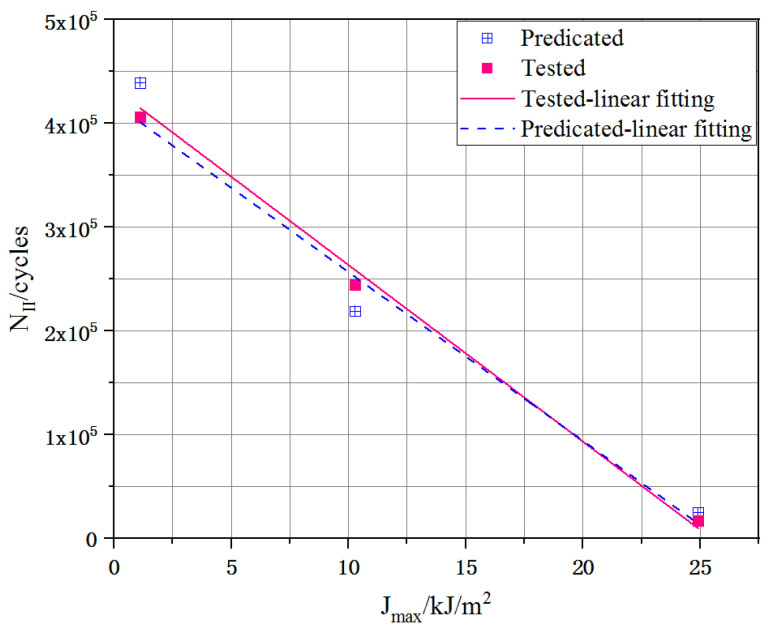
Comparison of the tested and predicated results of fatigue life during stages II.

**Figure 14 materials-16-00296-f014:**
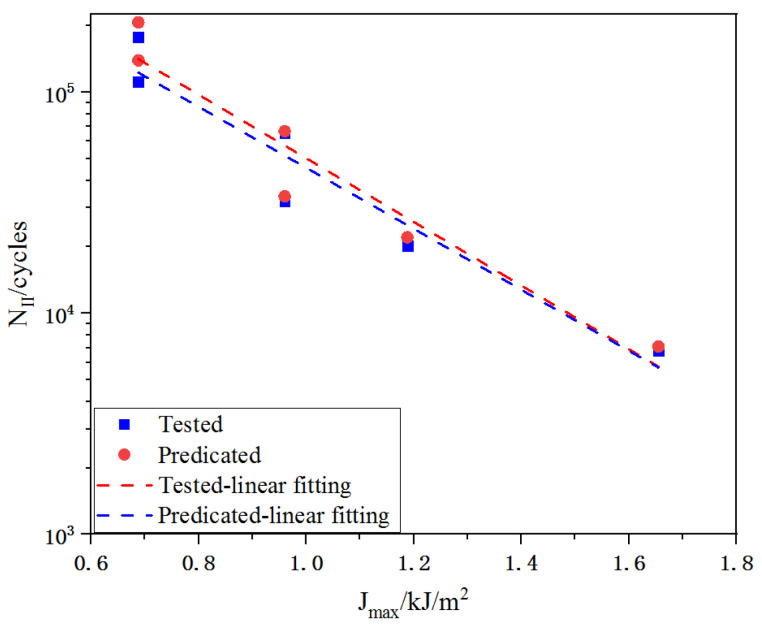
Comparison of tested and predicated fatigue life in stage II [29].

**Table 1 materials-16-00296-t001:** Chemical compositions of the cementitious materials (Unit [%]).

Cementitious Materials	SiO_2_	Al_2_O_3_	Fe_2_O_3_	CaO	MgO	K_2_O	Na_2_O	SO_3_	Others	LOI
Cement	21.60	4.35	2.95	63.81	1.76	0.51	0.16	2.06	1.61	1.19
Silica fume	98.07	-	0.12	0.51	0.31	0.53	0.14	0.12	0.19	0.01

Note: LOI, loss on ignition.

**Table 2 materials-16-00296-t002:** Physical properties of the steel fibers.

$\mathrm{Length},\;L_{f}\;(\mathrm{mm})$	$\mathrm{Diameter},\;d_{f}\;(\mathrm{mm})$	Aspect Ratio $\left(L_{f}/d_{f}\right)$	Density $\left(g/{\mathrm{cm}}^{3}\right)$	Tensile Strength (MPa)	Elastic Modulus (GPa)
6/13/20	0.2	30/65/100	7.8	2500	200

**Table 3 materials-16-00296-t003:** Mix proportion of the UHPC mixture with a fiber volume fraction of 2.0% (kg/m^3^).

Name	Cement	Silica Fume	Fine Aggregates		Superplasticizer	Water	Fiber Lengths
			0.16–0.31 mm	0.63–1.25 mm			
SF6	770	230	300	700	35	135.5	6 mm
SF13							13 mm
SF20							20 mm

**Table 4 materials-16-00296-t004:** Summary of the experimental calibration constants for Equation (1).

Series	τ	b	$m_{1}$	c	$R^{2}$
SF6	7.65	200	2.50 × $10^{-4}$	8.42	0.96
SF13	2.52		1.91 × $10^{-13}$	5.73 × $10^{-8}$	0.94
SF20	2.33		6.07 × $10^{-16}$	5.55 × $10^{-11}$	0.90

**Table 5 materials-16-00296-t005:** Summary of the experimental data of UHTCC under flexural fatigue loading.

Series	Fatigue Life		Crack Area			$J_{\max}$ $\;(\mathrm{kJ}/m^{2})$
	N	$N_{\mathrm{II}}$	$A_{I}$ $\;({\mathrm{cm}}^{2})$	$A_{\mathrm{II}}$ $\;({\mathrm{cm}}^{2})$	$A_{\mathrm{II}}$ $\;({\mathrm{cm}}^{2})$	
F-1	8501	6800	1.51	2.02	0.51	1.655
F-2	25,301	20,240	1.08	1.76	0.68	1.188
F-3-1	40,285	32,228	1.10	2.71	1.61	0.960
F-3-2	81,586	65,269	1.29	2.56	1.27	0.960
F-4	139,265	111,412	0.72	1.29	0.57	0.688
F-5	222,657	172,657	1.71	2.72	1.01	0.688

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
