# Peer review of "Application of the J-Integral and Digital Image Correlation (DIC) to Determination of Multiple Crack Propagation Law of UHPC under Flexural Cyclic Loading"

_materials, 2022, doi:10.3390/ma16010296_

Round 1
Reviewer 1 Report
The present manuscript reports the influence of steel fiber length on the fatigue crack propagation of a UHPC under flexural cyclic loading. After some revisions and the addition of a technical review, the manuscript may deserve to publish in this journal.
-. Spell out the full term at its first mention, and indicate its abbreviation in parentheses such as COD and DIC.
-. Add legends to the plot of Fig. 1. Please explain the results of the plots and the meaning of the results.
-. Please describe the images in Fig. 2. Please insert the scale bar.
-. The section number of 3.2.3 is duplicated in the section of “3. Results and discussion”.
-. There is no caption for Table 5.
Author Response
Reviewer #1:
The present manuscript reports the influence of steel fiber length on the fatigue crack propagation of a UHPC under flexural cyclic loading. After some revisions and the addition of a technical review, the manuscript may deserve to publish in this journal.
Comment 1: Spell out the full term at its first mention, and indicate its abbreviation in parentheses such as COD and DIC.
Reply: we are appreciative of the reviewer’s suggestion. The full term of the abbreviations has been revised at its first mention.
Comment 2: Add legends to the plot of Fig. 1. Please explain the results of the plots and the meaning of the results.
Reply: Thanks for this suggestion. The legends have been added to the plot of Fig.1. Due to the mix proportion design of the UHPC is based on the particle dense packing theory, it is important to quantitative evaluate the Particle size distribution of the cementitious materials.
Comment 3: Please describe the images in Fig. 2. Please insert the scale bar.
Reply: we are appreciative of the reviewer’s suggestion. Fig.2. intuitively describe the distribution pattern for the steel fiber in the UHPC. It can be seen that the longer fiber direction distribution tends to be perpendicular to the flexural loading, which indicated that the longer steel fiber has an important effect on the fiber bridging to inhibit the crack propagation behavior. The scale bar has been inserted in the Fig.2.
Comment 4: The section number of 3.2.3 is duplicated in the section of “3. Results and discussion”.
Reply: This comment is constructive. The section number of 3.2.3 in the section of “3. Results and discussion” have been revised in the manuscript and other section numbers have also been revised.
Comment 5: There is no caption for Table 5.
Reply: The caption for the Table 5 has been added in the revised version.
Reviewer 2 Report
Dear Murray Tong
Editor of Materials Journal
The manuscript under the title of "Application of the J-integral and 3D-DIC to the determination of multiple crack propagation law of UHPC under flexural cyclic loading" is about the fatigue behavior of UHPC with straight Steel fibers The prediction model of fatigue behavior is presented, but the type of sample is limited and only one type of concrete and three different lengths of fibers have been used.
The level of originality of novelty is average, and similar works have been published.
The strength of the manuscript is the interpretation of the results.
Yours sincerely,
Author Response
Reviewer #2: The manuscript under the title of "Application of the J-integral and 3D-DIC to the determination of multiple crack propagation law of UHPC under flexural cyclic loading" is about the fatigue behavior of UHPC with straight Steel fibers The prediction model of fatigue behavior is presented, but the type of sample is limited and only one type of concrete and three different lengths of fibers have been used.
The level of originality of novelty is average, and similar works have been published.
The strength of the manuscript is the interpretation of the results.
Reply: This comment is appreciated. The novelty of this manuscript is that employing the multiple crack covering areas and fatigue J-integral amplitudes to quantitatively evaluate the fatigue crack propagation rate and predicate the fatigue life of the UHPC during the steady development stage. Although this paper has many deficiencies, we firmly believe that the novelty will be more prominent with the development of this research.
Reviewer 3 Report
The manuscript described the fatigue crack propagation behavior of ultra-high-performance concrete (UHPC) incorporated with different steel fiber lengths under flexural cyclic loading. The Authors based on the Paris law and nonlinear fracture mechanics. The Authors concluded that it is reasonable to predict the fatigue life of the UHPC based on the J-integral according to the DIC technique. The paper potentially contributes to the literature as it presents novel results of interest for practice engineering purposes.
General report and comments:
· Table 1. Please add a measure of the quantity of components. Unit [%] – percentage?
· Page 19, line 383. The Table is without description. Please add a legend.
· Conclusion chapter. Additionally, there should be closing remarks after the general conclusions (after the bullet points of conclusions), keeping in mind all the outcomes obtained.
Author Response
Reviewer #3:
The manuscript described the fatigue crack propagation behavior of ultra-high-performance concrete (UHPC) incorporated with different steel fiber lengths under flexural cyclic loading. The Authors based on the Paris law and nonlinear fracture mechanics. The Authors concluded that it is reasonable to predict the fatigue life of the UHPC based on the J-integral according to the DIC technique. The paper potentially contributes to the literature as it presents novel results of interest for practice engineering purposes.
Comment 1: Table 1. Please add a measure of the quantity of components. Unit [%] – percentage?
Reply: Thank you for your comments. The measure of the quantity of components has been added in the revised version.
Comment 2: Page 19, line 383. The Table is without description. Please add a legend.
Reply: The caption for the Table 5 has been added in the revised version.
Comment 3: Conclusion chapter. Additionally, there should be closing remarks after the general conclusions (after the bullet points of conclusions), keeping in mind all the outcomes obtained.
Reply: Thank you for your constructive comment. The conclusion section of this manuscript has been revised in the revised version.